# PROGRESSIVE FUSION FOR MULTIMODAL INTEGRATION

## ABSTRACT

Integration of multimodal information from various sources has been shown to boost the performance of machine learning models and thus has received increased attention in recent years. Often such models use deep modality-specific networks to obtain unimodal features, which are combined to obtain "late-fusion" representations. However, these designs run the risk of information loss in the respective unimodal pipelines. On the other hand, "early-fusion" methodologies, which combine features early, suffer from the problems associated with feature heterogeneity and high sample complexity. In this work, we present an iterative representation refinement approach called Progressive Fusion, a model-agnostic technique that makes late stage fused representations available to early layers through backward connections, improving the expressiveness of the representations. Progressive Fusion avoids the information loss which occurs when late fusion is used while retaining the advantages of late fusion designs. We test Progressive Fusion on tasks including affective sentiment detection, multimedia analysis, and time series fusion with different models, demonstrating its versatility. We show that our approach consistently improves performance, for instance, attaining a 5% reduction in MSE and 40% improvement in robustness on multimodal time series prediction.

## 1 INTRODUCTION

Traditionally, research in machine learning has focused on different sensory modalities in isolation, though it is well-recognized that human perception relies on the integration of information from multiple sensory modalities. Multimodal fusion research aims to fill this gap by integrating different unimodal representations into a unified common representation (Turchet et al., 2018; Baltrušaitis et al., 2018).

Typically, fusion techniques fall into two categories, *early fusion* and *late fusion*, depending on where the information from each modality is integrated into the feature pipeline (Varshney, 1997; Ramachandram and Taylor, 2017). While theoretically, early fusion models tend to be more expressive, in practice, they are more commonly used for homogeneous or similar modalities (Ramachandram and Taylor, 2017). On the other hand, late fusion models are more effective in combining diverse modalities. This has generally been attributed to the challenges like feature shifts, cross-modal distributional changes, differences in dimensionality, etc., when dealing with heterogeneities across diverse modalities such as text and image (Mogadala et al., 2021; Yan et al., 2021).

In this work, we aim to bridge this divide by using backward connections which connect the late fused representation ( à la late fusion) to unimodal feature generators, thus providing cross-modal information to the early layers ( à la early fusion). This creates a model that learns to progressively refine the fused multimodal representations.

We show that our proposed technique called progressive-fusion (Pro-Fusion) results in improvements of different multimodal fusion architectures, including recent *state of the art models* such as MAGXLNET (Rahman et al., 2020), MIM (Han et al., 2021) and MFAS (Pérez-Rúa et al., 2019). Our experiments show that training with the Pro-Fusion design results in more accurate and robust models than baseline state-of-the-art architectures.

**Contributions:** (1) We propose a framework to bridge the gap between early and late fusion via backward connections. (2) We apply this model-agnostic approach to a broad range of state of the

art models for a diverse set of tasks. (3) We show, through rigorous experiments, that models trained with Pro-Fusion are not just consistently more accurate, but also considerably more robust than the corresponding standard baseline models. We show up to 2% improvement in accuracy over state of the art sentiment prediction models and up to 5% reduction in MSE and 40% improvement in robustness on a challenging multimodal timeseries prediction task.

## 2 BACKGROUND AND RELATED WORK

### 2.1 MULTIMODAL FUSION

Multimodal learning is a specific type of supervised learning problem with different types of input modalities. We are provided with a dataset of $N$ observations $\mathcal{D} = (X^j, Y^j)_{j=1}^N$, where all $X^j$ come from a space $\mathcal{X}$ and $Y^j$ from $\mathcal{Y}$, and a loss function $L : \mathcal{Y} \times \mathcal{Y} \to \mathbb{R}$ which is the task loss. Our goal is to learn a parametric function $\mathcal{F} : \mathcal{X} \to \mathcal{Y}$ such that the total loss $\mathcal{L} = \sum_j L(\mathcal{F}(X^j), Y^j)$ is minimized. In multimodal fusion the space of inputs $\mathcal{X}$ naturally decomposes into $K$ different modalities $\mathcal{X} = \prod_{i=1}^K \mathcal{X}_i$. Correspondingly any observation $X^j$ also decomposes into modality specific components $X_i^j$ i.e. $X^j = (X_1^j, X_2^j, \ldots X_K^j)$.

A natural way to learn such a function with a multimodal input is to have an *embedding component* which fuses information into a high dimensional vector in $\mathbb{R}^d$, where $d$ is the size of the embedding, and a *predictive component* $P$ which maps the embedding vector from $\mathbb{R}^d$ to $\mathcal{Y}$. Furthermore, since different modalities are often of distinct nature and cannot be processed by similar networks (e.g. text and image), the embedding generator is decomposed into (a) unimodal feature generators $G_i : \mathcal{X}_i \to \mathbb{R}^{d_i}$ which are specifically designed for each individual modality $\mathcal{X}_i$ and (b) a fusion component $F : \prod_i \mathbb{R}^{d_i} \to \mathbb{R}^d$ which fuses information from each individual unimodal vector. $F$ is provided with unimodal representations of the input $X^j$ obtained through embedding networks $G_i$. The unimodal feature generators $G_i$ can have different kinds of layers including 2D convolution, 3D convolution and fully connected layers. $F$ is the layer where the embeddings obtained from different modalities are fused. $F$ is called the *fusion* or *shared representation* layer. $F$ has to capture both unimodal dependencies (i.e. relations between features that span only one modality) and multimodal dependencies (i.e. relationships between features across multiple modalities).

### 2.2 PRIOR APPROACHES TO FUSION

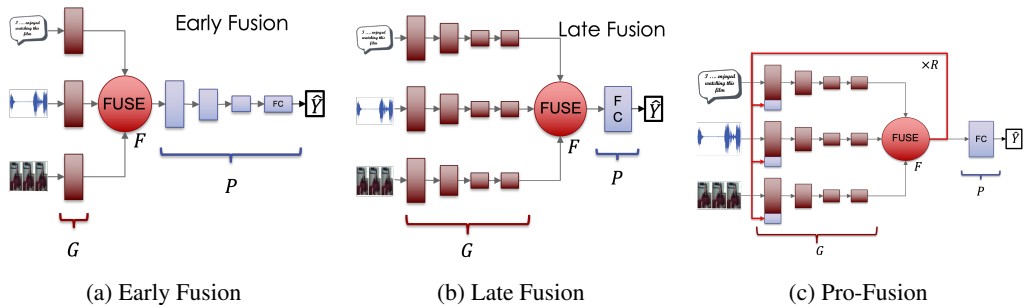

(a) Early Fusion       (b) Late Fusion       (c) Pro-Fusion

Figure 1: Representative Multimodal Fusion Architectures of a) Early fusion , b) Late fusion and c) Pro-Fusion. We have also indicated the components mentioned in Section 2.1 viz. the unimodal feature generators $G$, fusion layer $F$ and predictive network $P$ in the figures. Generally models with high capacity $P/G$ are considered early/late fusion respectively. The key difference between a late fusion architecture and pro-fusion architecture are the skip-back connections, indicated in red.

Many recent works including that of Vielzeuf et al. (2018), Sankaran et al. (2021), Pérez-Rúa et al. (2019), Hazarika et al. (2020) design new deep architectures. Vielzeuf et al. (2018) proposed a CentralNet design based on aggregative multi-task learning. Sankaran et al. (2021) design a Refiner Fusion Network (Refnet) trained via cyclic losses. Pérez-Rúa et al. (2019) used neural architecture search to find a good architecture for convolutional networks. Hsu and Glass (2018) and Khattar et al. (2019) use multimodal autoencoders to learn better representations. Tsai et al. (2019b)

improved upon the factor model based approach of Hsu and Glass (2018). Nagrani et al. (2021) modify the multimodal transformer (Tsai et al., 2019a) to incorporate bottlenecks.

Our proposed method, though technically an architecture change, is a single change that *treats the existing model as given*. It is closer in spirit to a black-box change, compared to the aforementioned methods. Hence it is *complementary* to this line of work. We experiment with many of the aforementioned models to show how our proposal consistently improves performance.

**Fusion Techniques** Other than basic fusion layers such as pooling and concatenation, other common layers used include aggregation (Khan et al., 2012), tensor factorisation (Liu et al., 2018; Zadeh et al., 2017), attention (Tsai et al., 2019a) and memory modules (Zadeh et al., 2018a). Rahman et al. (2020) design a model using pre-trained transformer to achieve state of the art results on the multimodal sentiment benchmarks. These works propose specific fusion techniques, they design specific forms of the $F$ function (see Figure 1). Our proposed technique is *agnostic to the choice of the fusion function $F$* and is *orthogonal* to these ideas.

**Model Agnostic Methods** Model independent methods to improve fusion by using train-time objectives based on mutual information (Colombo et al., 2021; Bramon et al., 2011) or contrastive estimation (Liu et al., 2021) have been widely explored. Our proposal is distinct from these methods in that it adds backprojective connections. These model-agnostic proposals are generally orthogonal to our approach, and potentially can be combined *to achieve further improvements*. For example, in our experiments we will show that our method can increase performance on the model-agnostic GB (Wang et al., 2020a) based approaches as well.

## 3 PROGRESSIVE FUSION (PRO-FUSION)

### 3.1 MOTIVATING EXAMPLE

Consider the task of determining the location of an entity from video and text. For instance, suppose the system has to detect the coordinates, in a given image, of an object specified through a textual command. For the image of the dog provided in Figure 2, the text might be 'find the tennis ball' or 'find the blue bone'. The task is not solvable using a single modality, as the image only contains the objects and their location, whereas the text only mentions the object of interest.

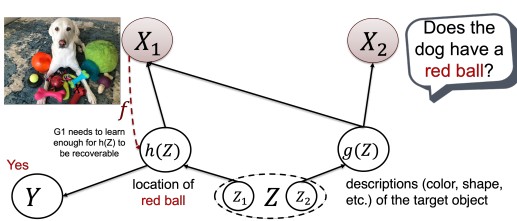

Figure 2: Motivating example. The target corresponds to the location in the image of the object described in the audio modality (dog, ball, bone etc). Also shown is the generative model where $Z$ is the latent vector that determines the outcome $Y$ via $h(Z)$. $g(Z)$ is independent of $Y$ given Z. $X_1$ is a combination of $h(Z)$ and $g(Z)$.
.

Consider what might happen with a late-fusion scheme. A significant part of the representation capacity of the image features might be devoted to capturing the dog, the bone, the carpet etc. Hence, determining the red ball's coordinates will be more challenging unless the image feature generator has access to the textual information. More generally, if the unimodal feature generators are bottlenecked or not powerful enough, the required information to predict the output might be lost or compressed too much to be recovered correctly. With early fusion, the image feature generator knows which object to focus on and can be directed toward the relevant information, namely the red ball.

Figure 2 also shows an abstract graphical model for this situation. $X_1$ represents the entire input image, while $Z$ represents an abstract state of the environment (with objects and coordinates). The output $Y$ (e.g., coordinate target) is determined by the feature function $h$, so $Y \leftarrow h(z)$ ( i.e., $h(Z)$ contains sufficient statistics about the location of the object). The information about these features is present in $X_1$ (obtained by applying unknown function $f$ to $h(Z)$); however, $X_1$ has nuisance variables (e.g., other objects) or a corrupted version of $h(z)$. $g(Z)$ represents descriptions like colour, shape, etc. of the target object. The location $h(Z)$ and visual characters $g(Z)$ combined

form (part of) the image. In this case, $Y$ may not be identifiable purely via $X_1$ . For the image example, this is because $X_1$ has not just the target but other objects, which means that without characterizing the desired target, a specific location cannot be specified. But in the presence of input $X_2$ (in this case text), the target is identifiable *even* if $X_2$ by itself is not informative about $Y$ . [1]

In such a scenario, with a late fusion based approach, if the encoder $G_1$ (unimodal feature generator for mode $\mathcal{X}_1$) is not sufficiently expressive, the overall networks may not be able to learn the perfect function $f$ even in the presence of modality $\mathcal{X}_2$. Such learning failure can happen during late fusion when the network $F_1$ has already pooled together $h$ and $g$ in a non-invertible manner. On the other hand, if the features from $X_2$ were made available to the feature encoder for $X_1$, it can learn to ignore or mask nuisance variation/corruption. Access to those $X_2$ features requires the model to perform some form of early fusion. However, the corresponding early integration is challenging if the underlying modalities $\mathcal{X}_2$ and $\mathcal{X}_1$ are very different.

More generally, fusion at higher-level features runs into a "fuse it or lose it" situation where relevant information – especially conditionally relevant information – that is not fused by the fusion layer is at risk of being lost. From the motivating example, only in the presence of $X_2$ (speech command) could we process $X_1$ (image) to get $h(Z)$ (location). The problem becomes intractable if location information in $X_1$ is corrupted before the fusion layer. This is happening because the unimodal feature generation is unaware of features coming from other modalities. Early fusion does not face this problem but cannot handle heterogeneous modalities well, requiring many parameters. This leads us to our basic problem: designing a generic approach combining late and early fusion advantages. To this end, we propose a model-agnostic scheme that provides late-stage multi-modal fusion features to the early stages of unimodal feature generators.

## 3.2 PRO-FUSION

We build a scheme based on backprojective connections which can be applied to any given base architecture. Our scheme considers any given base design as a single step of an iterative process. The base design is augmented to take an additional context vector as input, which serves to provide information from 'late' fused representations. At each iteration, the current output representations of the base model are provided via the context vector as an additional input for the next step. More formally, given a base model $\mathcal{F}$ with input $x = (x_i, x_2, ..x_k)$, we want to create an augmented model $\hat{\mathcal{F}} : \mathcal{X} \times \mathbb{R}^d \to \mathcal{Y}$ with additional input $c \in \mathbb{R}^d$ such that $c = 0 \implies \hat{\mathcal{F}}(x, c) = \mathcal{F}(x)$. Recall that the function $\mathcal{F}$ mentioned in Section 2.1 is given by $\mathcal{F}(x) = P(F(G_1(x_1), G_2(x_2), ..G_K(x_K)))$.

We create the desired network $\hat{\mathcal{F}}$ by providing $c$ to the unimodal feature generators $G_j$. We use the output of the fusion layer $F$ and project it back into the network as $c_t$ via the matrix/function $W_i$. This creates an iterative network which we run for $R$ steps. The final vector $c_R$ after $R$ steps serves as the output of fusion which is then provided to the predictor model $P$.

The ability to send information backward in the network addresses the problem raised earlier in Section 3.1 3. The encoder $G_1$ for $X_1$ can now gain a handle on $g(Z)$ via the fused output $c_1$. Specifically if the model can compute $g(z)$ from $W(c_1)$, then in the second iteration step, one can recover from $X_1$ the real value of $h(Z)$, which then directly determines the target $Y$. On the other hand if $X_2$ is not useful or if $G_1$ cannot process the fused vector efficiently, then $W(.)$ can be zeroed out and the overall model is no worse than the baseline model. We also present in the Appendix E, some mathematical analysis as to the representation power of our approach.

The importance of multimodal backward connections can also be interpreted from the perspective of the graphical model in Figure 7. A standard message passing routine (Koller and Friedman, 2009) on the aforementioned graph, will have the message from $X_2$ effecting the belief of target $Y$ via two paths: a) one along $X_2, g(Z), Z, h(Z)$ and the other along $X_2, g(Z), X_1, h(Z)$. Notice that along this second path, message from the modality $X_2$ is received at $X_1$ before further processing. This path makes features from modality $X_2$ available to the feature generator of $X_1$, which is exactly what our backprojection layer accomplishes. A caveat is that unlike this example, in general we do not know which way to route the messages (as the dependence graph maybe unknown). As such in our proposal we treat all modalities symmetrically and re-cycle information through all of them.

---

[1]For an example in terms of equations refer to the Appendix B.1

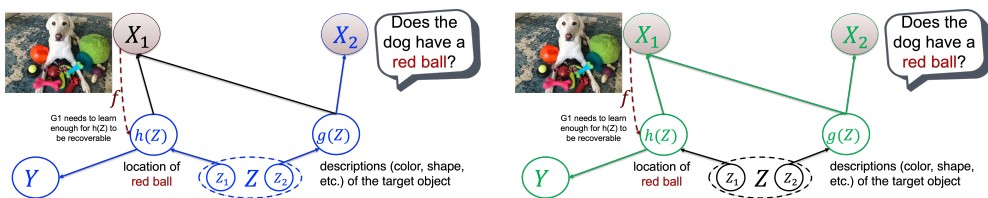

Figure 3: A standard message passing routine on the graph for the motivating example will have the message from $X_2$ affecting the belief of target $Y$ via two paths. Late fusion is only guaranteed to cover the path outlined in blue (left), which does not include $X_1$, potentially resulting in information loss. Progressive Fusion ensures that the path outlined in green (right) will also be covered, making features from modality $X_2$ available to the feature generator of $X_1$, thus preventing information loss.

An astute reader might notice similarities with deep unfolding networks (Balatsoukas-Stimming and Studer, 2019; Hershey et al., 2014). However, these are not designed for multimodal data, nor adapted to it, to the best of our knowledge. In contrast, ProFusion was specifically designed to solve a problem in multimodal data fusion: the "fuse it or lose it" situation. *Deep unfolding/iterative models that do not cycle cross-modal information still suffer from the "fuse it or lose it" problem.* This was confirmed by our experiments where we show that ProFusion provides additional improvement over deep unrolling style iterative models. Secondly, unrolling is just one method to train the backward connections. We refer the readers to Appendix A, for an expanded discussion on this.

## 4 EXPERIMENTS

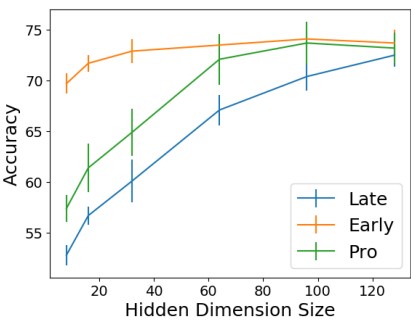

Figure 4: Accuracy of late, early and pro-fusion models over varying levels of inner dimension. Each point corresponds to the performance of the model when the hidden dimension is set to the values of $d$ on the x axis.

In this section, we empirically show that Pro-Fusion improves performance of multimodal deep learning SOTA architectures on a variety of tasks. First we verify our intuition for the advantage of backward connections in a synthetic experiment. Next, we experiment with datasets in sentiment prediction (Zadeh et al., 2018b), multimedia classification (Vielzeuf et al., 2018) and financial timeseries prediction (Sardelich and Manandhar, 2018). We also explore how our approach affects robustness for noisy time series data. Finally we evaluate the impact of varying the number of unrolling steps and analyze how the model performance as well as unimodal representations evolve. For all the datasets we use SOTA and near-SOTA models, while keeping a diversity of fusion techniques and network designs. For each dataset and architecture combination, we either use established hyperparameters and/or choose the best hyperparameter from our own experiments. Next, for the same architecture, we *add backward connections* from the fusion layer output and train with the exact same hyperparameters. We *do not perform any hyperparameter tuning for our modified models*, so the reported results are a lower bound the Pro-Fusion performance. We opt for this process to isolate the effects of adding backward connections from those of tuning hyperparameters.

### 4.1 SYNTHETIC DATASET

To verify the intuition described in the 'Motivating Example' Section, we first create a synthetic experiment. For this purpose we encode a smooth random function in modality $\mathcal{X}_1 \subset \mathbb{R}^D$. Specifically the $d^{\text{th}}$ component of $X_1$ has the value of the function at the point $d/D$. Next, in modality $\mathcal{X}_2$, we provide position embeddings of a randomly chosen lattice point $l \in \{0, 1/D, 2/D, ...1\}$. The output label $Y$ is the first non-zero digit of $x_l$. This is conceptually a simple task as one can infer the component from the modality $X_2$ and simply read on the corresponding component from $X_1$. How-

ever if the model is late fusion, where the input modalities might go through a lower dimensional representation; the specific values of each component in $X_1$ is lost, and the model cannot correctly predict the label. Note that in this case, each instance of $X_1$ contains a different function; because a fixed function might be directly learned by the network.

In Figure 4, we plot the accuracy of a 2 layer MLP trained on this task with different sizes for the hidden layer. The argument from Section 3.1 suggests that early fusion is more effective than late fusion when the hidden layers are smaller. It also suggests that the effect of progressive fusion is larger when the feature layers input to late fusion is smaller. This is confirmed by the experiments, where the gap between the pro-fusion model and late fusion model reduces as the size of the hidden representation increases. Finally, for a large enough hidden representation, the performance of late fusion matches that of early fusion. Additional analysis on synthetic data is shown in the Appendix.

## 4.2 MULTIMEDIA CLASSIFICATION

**Datasets.** We first evaluate our proposed design changes on AV-MNIST (Vielzeuf et al., 2018), a popular benchmark dataset used for multimodal fusion (Pérez-Rúa et al., 2019; Joze et al., 2020). It is an audio-visual dataset for a digit classification task. The data is prepared by pairing human utterances of digits obtained from FSDD dataset [2] with images of written digits from MNIST. This dataset has 55K training, 5K validation, and 10K testing examples. To prepare the dataset we use the processing stack of Cassell (2019). The preprocessing involves adding corruption to both modalities, so that no single modality is sufficient (Vielzeuf et al., 2018).

|       | Accuracy ↑ | |
|-------|------|------|
| Model | Base | Ours |
| LF    | 71.4 | **71.6** |
| LFN   | 71.1 | **71.8\*** |
| MFM   | 71.4 | **72.2\*** |
| GB    | 68.9 | **69.3** |
| Refnet| 70.6 | **71.2\*** |
| MFAS  | 72.1 | **72.5\*** |
| MBT   | **70.3** | **70.3** |

Table 1: Results on digit classification task with AVMNIST for various fusion architectures. The performance metric is Accuracy, and was measured on five trials. Our method *outperforms the baseline in almost all instances*. Scores above 1 standard deviation of the base models, indicating significance, have been marked with a *.

**Models.** **LF** is the baseline late fusion architecture used in Vielzeuf et al. (2018). **MFAS** is the architecture search based model used by Pérez-Rúa et al. (2019). It is the current SOTA on AV-MNIST. The exact architecture is presented in the Appendix D.3. We use the model obtained by search and add the backward connections. **LFN** is the low rank tensor fusion approach (Zadeh et al., 2017) adapted to this dataset, while **MFM** refers to the factorization method of Tsai et al. (2019b) for learning multimodal representation. **GB** and **Refnet** are the gradient blending and refiner network based approaches of Wang et al. (2020a) and Sankaran et al. (2021) respectively. **MBT** is the multimodal transformer model of Nagrani et al. (2021).

Our results are presented in Table 1. Amongst all the methods we evaluated, Pro-MFAS was the best model and beats its standard counterpart by 0.4 accuracy points. We also observe similar improvements in using Pro-Fusion with the MFM design. In fact the Pro-fusion MFM model was competitive with the current state of the art MFAS model. Meanwhile, the gradient blending (GB) fusion approach seems to not generalize on this dataset and performs worse than even late fusio.

## 4.3 SENTIMENT PREDICTION

**Datasets.** We empirically evaluate our methods on two datasets CMU-MOSI (Wöllmer et al., 2013) and CMU-MOSEI (Zadeh et al., 2018b). CMU-MOSI is sentiment prediction tasks on a set of short youtube video clips. CMU-MOSEI is a similar dataset consisting of around 23k review videos taken from YouTube. Both of these are used generally for multimodal sentiment analysis experiments. Audio, video, and language modalities are available in each dataset.

**Models.** **FLSTM** is the early fusion type baseline LSTM architecture used by Zadeh et al. (2017), while **LFN** is the low rank tensor representation of model of Zadeh et al. (2017). multimodal features. (Hazarika et al., 2020). **MAGBERT** and **MAGXLNET** (Rahman et al., 2020) are BERT (Devlin et al., 2018) based state of the art models on these datasets. These architectures use a gating

---

[2]https://www.tensorflow.org/datasets/catalog/spoken_digit

|        | $Acc_7 \uparrow$ | | $Acc_2 \uparrow$ | |
|--------|------|------|------|------|
| Model  | Base | Ours | Base | Ours |
| FLSTM    | 31.2 | **31.8** | 75.9 | **76.8** |
| LFN      | 31.2 | **32.1** | 76.6 | **77.2** |
| MAGBERT  | 40.2 | **40.8** | 83.7 | **84.1** |
| MAGXLNET | 43.1 | **43.5** | 85.2 | 85.5 |
| MIM      | 45.5 | **46.3** | 81.7 | **83.4** |

Table 2: Results on sentiment analysis on CMU-MOSI. $Acc_7$ and $Acc_2$ denote accuracy on 7 and 2 classes respectively. Boldface denotes statistical significance.

|        | $Acc_7 \uparrow$ | | $Acc_2 \uparrow$ | |
|--------|------|------|------|------|
| Model  | Base | Ours | Base | Ours |
| FLSTM    | 44.1 | **44.8** | 75.1 | **75.8** |
| LFN      | 44.9 | **46.1** | 75.3 | **76.4** |
| MAGBERT  | 46.9 | 47.1 | 83.1 | **83.6** |
| MAGXLNET | 46.7 | 47.1 | 83.9 | 84.2 |
| MIM      | 53.3 | **54.1** | 79.1 | **80.1** |

Table 3: Results on sentiment analysis on CMU-MOSEI. $Acc_7$ and $Acc_2$ denote accuracy on 7 and 2 classes respectively. Boldface denotes statistical significance.

mechanism (Wang et al., 2019) to augment a pretrained transformer. **MIM** (Han et al., 2021) is a recent near-SOTA architecture. It combines BERT text embeddings with modality specific LSTMs.

We evaluate our change on the aforementioned five models on four metrics commonly used in the literature (Zadeh et al., 2017; Han et al., 2021). The binary and 7-class accuracy results are reported in Tables 2 and 3. We present the results of the remaining metrics (MAE and CORR) in Appendix 8. We observe consistent improvements in accuracy of non-transformer based models (FLSTM, LFM, MIM) ranging from 0.5% to 1.5%, while transformer based models improve by 0.3%. The comparatively smaller improvement in transformers could be due to the lack of additional information from other modalities when using BERT on text. For example, on CMU-MOSI, simply using BERT embeddings provides an accuracy of 78% which is higher than most non-BERT fusion models (Hazarika et al., 2020). Given the degree of sufficiency in the textual modality, performance is determined by the text network not by the fusion design.

### 4.4 FINANCIAL DATA

**Datasets.** We evaluate our approach on a multimodal financial time series prediction task (Sardelich and Manandhar, 2018). F&B, HEALTH, and TECH are prices and events related to publically listed companies organized according to the primary business sector. Within each sector, historical prices are used as time series inputs to predict the future price and volatility of a related stock. In this setting the different stocks in the same sector correspond to different modalities. Due to the significantly large number of available modalities, this task presents a different set of challenges (Emerson et al., 2019; Sardelich and Manandhar, 2018) than other datasets. Moreover, due to the inherently low signal-to-noise ratio in such time series, it presents a greater robustness challenge than other datasets (Liang et al., 2021a). On the other hand, due to the similar nature of the modalities this task is amenable to early fusion methods.

**Models.** We experiment with Transformers for time series (Sardelich and Manandhar, 2018) with both early fusion **EF transf** and late fusion **LF transf** variants. The other models we test are the multimodal fusion transformer **MulT** Tsai et al. (2019a), Gradient Blending **GB** approach from (Wang et al., 2020a). Finally as LSTMs are strong baselines on this task (Narayanan et al., 2019), we also use Early fusion **EFLSTM** and Late **LFLSTM** fusion LSTM models.

Because of the similar nature of the modalities, one might expect early fusion models to be effective. This can be seen in our results, where early fusion LSTM outperforms late fusion models. However, we note that, by using backward connections, the late fusion models, especially LFLSTM, become competitive with early fusion models. The nature of the dataset- low dimension time series with inherent noise- means we can also assess the models' robustness against modality corruption. We

| | Metric | MSE $\downarrow$ | | Robustness $\uparrow$ | |
|---|---|---|---|---|---|
| Model | Dataset | Base | Ours | Base | Ours |
| EFLSTM | F&B | 0.73 | **0.70** | 0.87 | **1.0** |
| | HEALTH | 0.308 | 0.306 | 0.54 | **0.83** |
| | TECH | 0.742 | **0.738** | 0.92 | 0.93 |
| LFLSTM | F&B | 0.77 | **0.73** | 0.74 | **0.83** |
| | HEALTH | 0.331 | **0.315** | 0.48 | **0.78** |
| | TECH | 0.736 | 0.737 | 0.96 | 0.96 |
| GB | F&B | 0.690 | 0.688 | 0.98 | 0.98 |
| | HEALTH | 0.318 | **0.305** | 0.67 | **1.0** |
| | TECH | 0.740 | **0.728** | 0.99 | 1.0 |
| LF Transf | F&B | 0.838 | **0.788** | 0.24 | **0.38** |
| | HEALTH | 0.337 | 0.331 | 0.34 | **0.46** |
| | TECH | 0.757 | 0.751 | 0.92 | 0.93 |
| MulT | F&B | 0.814 | **0.765** | 0.33 | **0.48** |
| | HEALTH | 0.333 | **0.329** | 0.0 | 0.08 |
| | TECH | 0.763 | **0.757** | 0.85 | 0.86 |
| EF Transf | F&B | 0.836 | **0.827** | 0.0 | 0.05 |
| | HEALTH | 0.335 | **0.326** | 0.45 | **0.63** |
| | TECH | 0.755 | **0.750** | 0.0 | 0.0 |

Table 4: Results on stock prediction on the three sectoral datasets. The performance is evaluated on the Mean Squared Error (MSE) metric evaluated on 10 trials. We also compute robustness metrics by testing on data corrupted with various noise levels and present the relative robustness scores. Scores which are outside the 1 standard deviation of the base model are highlighted.

add varying noise levels to the test data and see how the performance of the models changes with increasing noise. Following (Taori et al., 2020; Shankar et al., 2017; Liang et al., 2021a), the robustness of the model is assessed by computing the area under the performance vs. noise curve. Specifically, it is computed by discrete approximation of the following integral:

$$\tau = \int \text{MSE}(f, \sigma) - \text{MSE}(b, \sigma) d\sigma$$

where $\text{MSE}(., \sigma)$ is the MSE of the model on test-data with inputs corrupted with noise level $\sigma$. $f$ is the model the evaluated and $b$ is a baseline model. We choose late fusion transformer as our baseline, and scale the numbers between 0 and 1 [3]. From the results we can see that *Pro-Fusion provides greater improvements on late fusion compared to early fusion designs*. This suggests that the improvement is partly due to the backprojection acting as a bridge between early and late fusion.

### 4.5 ABLATION EXPERIMENTS

To assess the impact of multimodal backprojecting connections in the Pro-Fusion approach against vanilla iterative models, we conduct experiments on AVMNIST. We change the unimodal feature generators of the baseline models into an iterative model. Effectively, these models are similar to the Pro-Fusion model except that we connect the output features of the unimodal feature generators to their inputs instead of having multimodal connections (See Figure 13 in the Appendix). This allows us to distinguish between the effect of multimodal backprojection from the effect of generic iterative processing. We fixed the number of iterative steps to 2 (same as our Pro-Fusion models) and ran 8 trials for these alternate models, with the results reported in Table 5.

| | Accuracy $\uparrow$ | | |
|---|---|---|---|
| Model | Base | Ours | Iterative |
| LFN | 71.1 | **71.8** | 71.5 |
| MFM | 71.4 | **72.2** | 69.9 |
| GB | 68.9 | 69.3 | 69.2 |
| Refnet | 70.6 | **71.2** | 70.7 |

Table 5: Results on digit classification task with AVMNIST for various fusion architectures. The performance metric is Accuracy, measured on five trials.

The results indicate that, while iterative models do lead generally to some improvement over the baseline models, Pro-Fusion is still better. Moreover in some cases (such as MFM) iterative models

---

[3] For the unscaled value of the numbers refer to Appendix C

can be worse than the baseline. The key difference between a vanilla iterative model and Pro-Fusion is that Pro-Fusion allows unimodal feature generators access to information from other modalities. As such, unimodal feature generators can now produce features conditioned on the other modalities, while in the alternate approach, the unimodal features are blind to the features from other modalities.

We also run experiments to evaluate the effect of the dimensionality of the backprojecting connections. We adjust the dimensionality of the backprojecting connection $W$, up to 512 and evaluate multiple models on AVMNIST. One might expect that backprojections with very low dimensions will be similar to baseline models with no backward connection. On the other hand, with a high dimensionality in the backward connection, one runs into the same problem as early fusion of high parametric complexity. This expectation matches the empirical results, shown in Figure 5. We plot the accuracy (and standard error) of multiple models with varying backprojection sizes. Notice that, for comparability across models, we have normalized all curves by their respective baseline results.

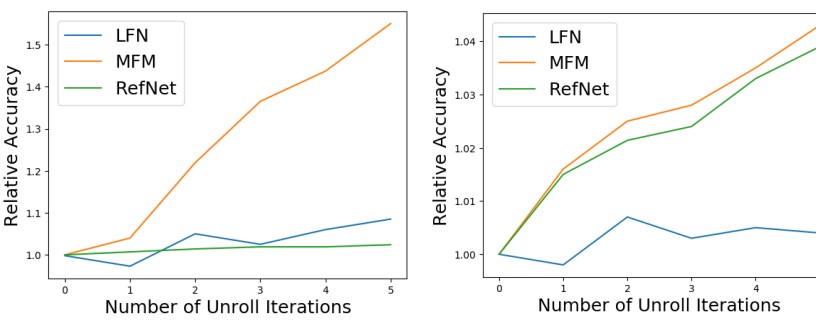

(a) Accuracy from audio representation     (b) Accuracy from image representation

Figure 6: Behavior of classifiers trained on the unimodal embedding against number of unrolling iterations. The lines plot the normalized accuracy of a linear model trained on input of fusion layer. We observe *increased accuracy* with more unrolling.

Next, we analyze how the unimodal representations evolve over the unrolling steps. For this purpose, we consider the activations of unimodal networks $\hat{G}_j$ (equivalently, the inputs for the late fusion layer) as the unimodal representations. For these of experiments, we use LFN, MFM and Refnet models on AVMNIST. We train a linear classifier based on the unimodal representations from the training data and find its accuracy on the test data.

In Figure 6 we plot the relative test accuracy of both the audio and image features against the iteration number for all the models. We can see gains in all models after one step of unrolling. Since we know that the modalities are incomplete/noisy (especially audio), the increasing accuracy can be attributed to additional information being available. This suggests that the unimodal modalities are integrating information from each other with more iterations.

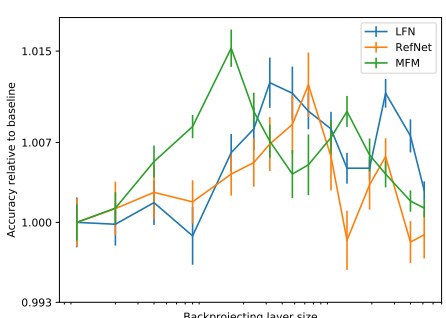

Figure 5: Relative accuracy of different models over varying dimensions of backprojecting connections. Each point corresponds to the normalized performance of the corresponding model when the hidden dimension is set to the values of on the x axis.

## 5 CONCLUSION

Our paper presents a model-agnostic approach to incorporate benefits of early fusion into late fusion networks via backward connections. We argued for some sufficient conditions when our backward connection based design to be more effective than usual fusion designs, supported by an artificial data experiment. With real data experiments, we make a case for using multimodal backward connections and show that Pro-fusion can improve even SOTA models.

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
