# A OVERVIEW OF FUSION TECHNIQUES AND RELATED WORKS

Multimodal fusion has been a heavily researched area for decades (Osadciw and Veeramachaneni, 2009; Varshney, 1997). Such models have been used for tasks ranging from video classification (Yang et al., 2016), action recognition (Cabrera-Quiros et al., 2019), and speech enhancement (Hou et al., 2017) to brain studies Sui et al. (2012), ecological applications (Taheri and Toygar, 2018), and monitoring systems (Varshney, 1997; Li and Seignez, 2018). However most models have primarily focused either on architectural changes or designing new fusion layers (Yan et al., 2021).

**Early Fusion** Figure 1a illustrates a general early fusion scheme. Early fusion, sometimes also called feature fusion in the early literature (Ayache et al., 2006; Chair and Varshney, 1986), creates a multimodal representation by combining unimodal information before they are processed. One can broadly interpret an Early Fusion scheme as one that integrates unimodal features before 'learning high level concepts'. Early fusion models have the ability to model highly complex dependencies between different modalities, however they generally face problems when dealing with heterogenous sources such as text and images.

**Late Fusion** In contrast to early fusion, late fusion designs learn 'high level semantic concepts' directly from unimodal features. Late fusion allows an easy way to aggregate information from diverse modalities and can easily incorporate pre-trained models (e.g. Rahman et al. (2020)). As such, late fusion has been the more commonly utilized framework for multimodal learning (Ramachandram and Taylor, 2017; Simonyan and Zisserman, 2015). However, late fusion models, illustrated in Figure 1b, run the risk of missing cross-modal interactions in the mixed feature space.

**Architecture Changes** Due to the wide variety of applications and tasks which require multimodal fusion, over the years a plethora of different architectures have been used. Some of the recent works include that of Vielzeuf et al. (2018), Sankaran et al. (2021), Pérez-Rúa et al. (2019), Hazarika et al. (2020), and Khan et al. (2012). Vielzeuf et al. (2018) proposed a multimodal fusion design called CentralNet that is based on aggregative multi-task learning. Sankaran et al. (2021) bring together ideas from CentralNet and Cycle-GAN (Zhu et al., 2017) and design a Refiner Fusion Network (Refnet). The Refnet design uses a de-fusion model trained via cyclic losses to align both unimodal and multimodal representations in a common latent space. Pérez-Rúa et al. (2019) used neural architecture search to find a good architecture for convolutional networks. The discovered architecture is a multistep fusion model that fuses information from different individual unimodal layers multiple times. Hsu and Glass (2018) and Khattar et al. (2019) used ideas from unsupervised learning to use multimodal autoencoders to learn better representations. Tsai et al. (2019b) improved upon the factor model based approach of Hsu and Glass (2018) by incorporating prior matching and discriminative losses. Nagrani et al. (2021) modify the multimodal transformer Tsai et al. (2019a) to incorporate bottlenecks.

Our proposed method, though technically an architecture change, is a single change that *treats the existing model as given*. It is closer in spirit to a black-box change, compared to the aforementioned methods. Hence it is *complementary* to this line of work. We experiment with many of the aforementioned models to show how our proposal consistently improves performance.

**Fusion Techniques** Other than basic fusion layers such as pooling and concatenation, other common layers use include aggregation (Khan et al., 2012), tensor factorisation (Liu et al., 2018), attention modules (Zadeh et al., 2018a; Tsai et al., 2019a), channel-swaps Wang et al. (2020b) and non-local gating (Hu et al., 2019; Wang et al., 2018; Liu et al., 2019). Rahman et al. (2020) used pre-trained transformer (Siriwardhana et al., 2020) along with Wang et al. (2019) modulation gate to achieve state of the art results on the multimodal sentiment benchmarks MOSI Wöllmer et al. (2013) and MOSEI Zadeh et al. (2018c). LFN (Zadeh et al., 2017) combined information via pooling projections of high dimensional tensor representation of multimodal features. These works propose specific fusion techniques, they design specific forms of the $F$ function (see Figure 1). Our proposed technique is *agnostic to the choice of the fusion function $F$* and thus is *orthogonal* to these ideas.

**Model Agnostic Methods** A number of alignment and information based losses have also been explored to improve fusion by inducing semantic relationships across the different unimodal repre-

sentations (Abavisani et al., 2019; Bramon et al., 2011; Liang et al., 2021b; Liu et al., 2021; Han et al., 2021) . These are purely train-time objectives and can be generally applied to most multimodal fusion models. Recently Wang et al. (2020a) proposed a new approach that can be applied to any multimodal architecture. Their approach called Gradient Blending (GB) tackles the problem of joint learning when different unimodal networks have varying capacity, by learning individual modality weights factors based on the model performances. Our proposal instead of adding losses or adding reweighing factors instead adds backprojective connections. So, these model-agnostic proposals are in general complementary to our approach, and can be *combined with it to achieve further improvements*.

**Memory based Fusion** Existing works using multiple fused representation computed over time, have been used for sentiment analysis (Gammulle et al., 2017; Zadeh et al., 2018a;b). The purpose of memory in those methods is retaining history for easier learning of interactions across time steps in a sequential input. On the other hand, in our proposal, context vector serves the purpose of making late-fusion features accessible to the unimodal network processing models/early stage features. Our proposal is entirely independent of any temporal axis/sequential nature in the input. Secondly, these methods capture historical relationships only in unimodal data, and memory is used directly over the concatenated unimodal features. On the other hand our approach provides multimodal information to unimodal feature generators.

**Deep Unfolding** Iterative neural networks (Chang et al., 2000; Hershey et al., 2014) have been successful for a variety of problems in computer vision such as inverse problems (Adler and Öktem, 2017; Chun et al., 2020), super resolution (Neshatpour et al., 2019) and other tasks (Chang et al., 2000; Balatsoukas-Stimming and Studer, 2019). Deep unfolding methods are specific recurrent models which at each iteration, pass the results of inference from previous iterations onward. Most such methods have been used for image super resolution (Zhang et al., 2020; Ning et al., 2020) and wireless communication systems (Balatsoukas-Stimming and Studer, 2019) While Pro-Fusion shares an unrolling design with these works, it differs from these methods in the following ways:

- Unrolling is used primarily as a way to train the backward connections, and are not the fundamental aspect of Pro-Fusion. ProFusion was specifically designed to solve a problem in multimodal data fusion: the "fuse it or lose it" situation; by adding cross-modal backward connections. In principle other methods such as equilibrium propagation (Ernoult et al., 2020), balance-tuning (Zhang et al., 2018) or other methods can also be used to train a self-iterative loop of Pro-Fusion models.

- Deep unfolding was not designed for multimodal data, nor adapted to it, to the best of our knowledge. Deep unfolding/iterative models that do not cycle cross-modal information still suffer from the "fuse it or lose it" problem. On the other hand, with ProFusion, unimodal representations adapt to multimodal features, as the model can, in future iterations, extract complementary information based on previous cross-modal features.

- Deep unfolding methods unroll a classical iterative algorithm such as "gradient descent" or "orthogonal message passing" and introduce trainable parameters for the update step. On the other hand, we consider a given multimodal fusion model as one step of the iteration.

- In our approach, parameters are shared between steps, which is not common in deep unfolding literature, where every iteration typically introduces a new set of parameters.

## B  FURTHER EXPERIMENTS

### B.1  SYNTHETIC EXPERIMENTS

We further explore the setting implied by the generative model described in Section 3.1. For this we generate data as from a generative model matching the dependencies in Figure 2. We set the function $h$ to be leaky-Relu and $g$ to be the sine function. We choose $f$ such that $f \circ h$ is linear. Note that the specific choice of $h, g$ makes the function $h + g$ non-invertible. $Z$ was sampled from a uniform distribution on [-2.5,2.5] and all linear transform matrix were also sampled from the standard normal distribution. The specific generative equations are presented in the equations below.

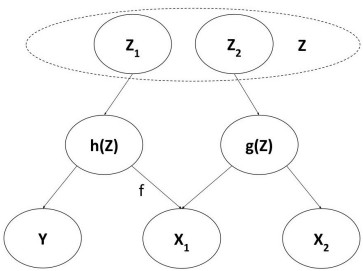

Figure 7

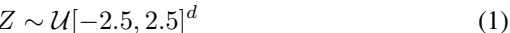

$$Z \sim \mathcal{U}[-2.5, 2.5]^d \tag{1}$$
$$W_1 \sim \mathcal{N}(0, I), W_2 \sim \mathcal{N}(0, I), W_y \sim \mathcal{N}(0, I) \tag{2}$$
$$X_1 = \text{lRelu}(W_1 Z) - 2 * |\eta| \sin(W_2 Z) + \varepsilon_1 \tag{3}$$
$$X_2 = \sin(W_2 Z) + \sigma_2 \varepsilon_2 \tag{4}$$
$$Y = W_y Z + \varepsilon_y \tag{5}$$
$$\tag{6}$$

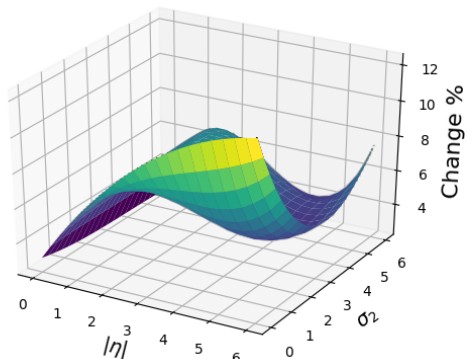

Figure 8: Percentage Improvement over varying levels of modality dependence. Note that this is percent improvement in MSE so higher is better

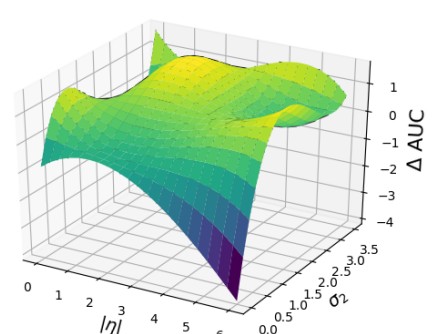

Figure 9: AUC change for pro-fusion model over normal fusion over varying levels of modality dependence. Since the metric is error, lower AUC is better

All noise terms $\varepsilon$ are sampled from a standard normal distribution. We train a 3 layer MLP to solve this regression task. Fusion was done via plain concatenation of the second layer. For the pro-fusion model, the same fusion vector was linearly transformed and fed along with the input. We varied the strength $\eta$ of corruption in $X_1$ and of $\sigma_2$ the noise in $X_2$, and ran for each such value 30 trials. In close to *90%* of all trials we found the pro-fusion model to perform better with an average improvement of *8%*. We plot the contours for multiple different runs in Figures 8.

We also do robustness evaluation of this model over different values of noise parameters. We use the AUC (area under ROC curve) metric for this purpose. In Figure 9 we plot the improvement in AUC of the pro-fusion model over direct fusion against the different values of $\eta/\sigma_2$

## B.2 EXPLORATORY EXPERIMENTS

In this section we explore various aspects of the backprojecting connections, such as the layer at which backprojecting connection joins, the number of unrolling steps and the inference complexity.

We measure the training time and inference time for pro-fusion with different architectures relative to that for the base model for different number of unrolling steps $R$. The straightforward way in which the current pro-fusion design uses the base model, suggests that both training and inference time should vary proportionately to the number of steps. This expectation is brought out in our experiments and can be seen in Figure 10b.

We also conduct experiments with determining at which upstream layer should the fused out should be connected back to. For this once again we run trials on the AVMNIST data. Our results are depicted in Figure 10a.

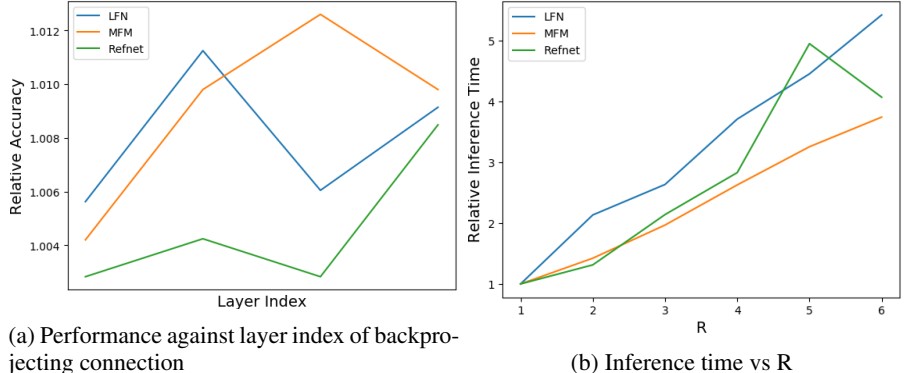

(a) Performance against layer index of backpro-
jecting connection

(b) Inference time vs R

## C FINANCIAL DATA DETAILS

The data itself is not licensed but following Liang et al. (2021a); Sardelich and Manandhar (2018) can be gathered from online records of historical stock prices and events.

- **FB** is composed of S&P 500 stocks which are part of food, meats and restaurant chains and includes the tickers CAG, CPB, DRI, GIS, HRL, HSY, K, MCD, MKC, SBUX, SJM, TSN, YUM.
- **HEALTH** is composed of following health-care and pharmaceutical sector tickers MRK, WST, CVS, MCK, ABT, UNH, TFX, PFE, GSK, NVS, WBA.
- **TECH** is composed of technology and information service stocks from NASDAQ. We include tickers AAPL, ADBE, AMD, AMZN, GOOG, HPQ, IBM, INTC, MSFT, MSI, NVDA, ORCL, QCOM, ZBRA.

### C.1 ROBUSTNESS COMPUTATION

We use the area under the performance-noise curve as the measure of robustness. This measure of robustness is the same as the one used by Liang et al. (2021a); Taori et al. (2020), and can be computed via the following integral:

$$\text{ROBUSTAUC} = \tau = \int Perf(f, \sigma) - Perf(b, \sigma) d\sigma$$

where $Perf(., \sigma)$ is the performance metric evaluated on a dataset corrupted with noise level $\sigma$, $f$ is the model to be evaluated and $b$ is a baseline model. Basically, the model is evaluated on the same dataset corrupted on an equally spaced grid of noise levels and the performance is averaged over all the noise configurations is used. Note that the $Perf$ as used by Shankar et al. (2017); Taori et al. (2020) is a positive metric like accuracy. For inverse metrics like MSE one has to use the negative of the above integral. For computing robustness we our experiments we use a transformer model as the baseline.

## D EXPERIMENTAL DETAILS

In this section we present the total results of all the experiments. We include metrics such as CORR (pearson correlation) which were not reported in the main body. We also report the average deviation of the scores in these tables.

### D.1 AUC MEASURES AND AVERAGE DEVIATION ON FINANCIAL DATA

Our results on financial time-series prediction, while qualitatively similar to results of (Liang et al., 2021a), is different because of using more number of target stocks and different time period. For completeness we also report the error on the dataset provided by them in Table 7.

| Model | | F&B MSE ↓ | F&B ROBUSTAUC ↑ | HEALTH MSE ↓ | HEALTH ROBUSTAUC ↑ | TECH MSE ↓ | TECH ROBUSTAUC ↑ |
|---|---|---|---|---|---|---|---|
| EFLSTM | Base | 0.73 (0.03) | 0.35 | 0.308 (0.005) | 0.018 | 0.742 (0.006) | 0.027 |
| | Our | 0.70 (0.04) | 0.40 | 0.306 (0.003) | 0.029 | 0.738 (0.004) | 0.028 |
| LFLSTM | Base | 0.77 (0.05) | 0.29 | 0.331 (0.009) | 0.016 | 0.736 (0.006) | 0.028 |
| | Our | 0.73 (0.05) | 0.33 | 0.315 (0.007) | 0.026 | 0.737 (0.005) | 0.028 |
| GB | Base | 0.690 (0.04) | 0.39 | 0.318 (0.03) | 0.022 | 0.740 (0.006) | 0.029 |
| | Our | 0.688 (0.02) | 0.39 | 0.305 (0.003) | 0.035 | 0.738 (0.005) | 0.029 |
| LF Transformer | Base | 0.838 (0.004) | 0.09 | 0.337 (0.004) | 0.012 | 0.757 (0.005) | 0.027 |
| | Our | 0.788 (0.004) | 0.15 | 0.331 (0.004) | 0.015 | 0.755 (0.005) | 0.028 |
| MulT | Base | 0.814 (0.005) | 0.13 | 0.333 (0.004) | 0.001 | 0.763 (0.005) | 0.025 |
| | Our | 0.765 (0.006) | 0.19 | 0.329 (0.005) | 0.002 | 0.757 (0.004) | 0.026 |
| EF transformer | Base | 0.836 (0.009) | 0. | 0.335 (0.001) | 0. | 0.755 (0.004) | 0. |
| | Our | 0.827 (0.009) | 0.01 | 0.326 (0.004) | 0.02 | 0.750 (0.004) | 0.00 |

Table 6: Results on stock prediction on the three sectoral datasets. The performance is evaluated on the Mean Sqaured Error (MSE) and ROBUSTAUC metric. Note we have already flipped the AUC sign for the inverse metric.

Table 7: Results on multimodal dataset of Liang et al. (2021a) in the finance domain

| Dataset Metric | F&B MSE ↓ | | HEALTH MSE ↓ | | TECH MSE ↓ | |
|---|---|---|---|---|---|---|
| | Base | Our | Base | Our | Base | Our |
| EF-LSTM | 1.836 | 1.753 | 0.521 | 0.511 | 0.119 | 0.124 |
| LF-LSTM | 1.891 | 1.786 | 0.545 | 0.522 | 0.120 | 0.121 |
| LF-Transformer | 2.157 | 2.112 | 0.572 | 0.566 | 0.143 | 0.144 |
| MulT | 2.056 | 2.032 | 0.554 | 0.553 | 0.135 | 0.132 |

## D.2 MULTIMEDIA

Complete results on AVMNIST along with the standard deviations of the performance are reported in Table 9

## D.3 HYPERPARAMETER DETAILS

For the AVMNIST dataset, we used LeNet style unimodal feature generators. For the image encoder we used a 4 layer network with filter sizes [5,3,3,3] and max-pooling with width of 2. For the audio encoder the networks was a 6 layer networks with filter sizes [5,3,3,3,3,3] and max-pooling of width 2. The channel width was doubled after each layer. For GB models, the validation size was 0.8 and the model is fine-tuned for gradient blending for 30 epochs. For the optimization process we tried random search on a logarithmic scale on the interval [1e-5, 5e-2]. We experimented with Adam, Adagrad, RMSProp, SGD optimizer with default configurations.

For the MFAS model, we did not do architecture search but instead used the final model presented by Pérez-Rúa et al. (2019). That model is shows in Figure 11. While we have tried to stay close to the method described in Pérez-Rúa et al. (2019); Liang et al. (2021a) for creation of this dataset, our version of AVMNIST is potentially different from the earlier reported results as no standard dataset is available. For financial time series prediction, we used 128 dimensional RNNs. For transformers we used a 3 layer network with 3 attention heads. The sequence length used for BPTT in all cases was 750. The optimization process was chosen in a similar way as mentioned previously.

**Parameter Sizes**

Models on MOSI/MOSEI, use and fine-tune BERT (or another similar large language model). The total number of trainable parameters for these models is >20M, and so the additional parameters introduced by ProFusion are especially small (relatively). In table 10 we present a comparison of the parameter size for the AVMNIST experiments.

| | $Acc_7 \uparrow$ | $Acc_2 \uparrow$ | MAE $\downarrow$ | CORR $\uparrow$ |
|---|---|---|---|---|
| | FLSTM | | | |
| Base | 31.2 (0.5) | 75.9 (0.5) | 1.01 | 0.64 |
| Our | **31.8** (0.4) | **76.8** (0.3) | 1.0 | 0.66 |
| | LFN | | | |
| Base | 31.2 (0.4) | 76.6 (0.4) | 1.01 | 0.62 |
| Our | **32.1** (0.6) | **77.2** (0.2) | 1.01 | 0.62 |
| | MAFBERT | | | |
| Base | 40.2 (0.4) | 83.7 (0.3) | 0.79 | 0.80 |
| Our | **40.8** (0.4) | **84.1** (0.3) | 0.79 | 0.80 |
| | MAGXLNET | | | |
| Base | 43.1 (0.2) | 85.2 (0.4) | 0.76 | 0.82 |
| Our | **43.5** (0.3) | 85.5 (0.2) | 0.76 | 0.83 |
| | MIM | | | |
| Base | 45.5 (0.1) | 81.7 (0.2) | 0.72 | 0.75 |
| Our | **46.3** (0.2) | **83.4** (0.5) | 0.71 | 0.77 |
| | $Acc_7 \uparrow$ | $Acc_2 \uparrow$ | MAE $\downarrow$ | CORR $\uparrow$ |
| | FLSTM | | | |
| Base | 44.1 (0.2) | 75.1 (0.3) | 0.72 | 0.51 |
| Our | **44.8** (0.5) | **75.8** (0.3) | 0.72 | 0.52 |
| | LFN | | | |
| Base | 44.9 (0.3) | 75.3 (0.4) | 0.72 | 0.52 |
| Our | **46.1** (0.3) | **76.4** (0.3) | 0.71 | 0.52 |
| | MAFBERT | | | |
| Base | 46.9 (0.7) | 83.1 (0.4) | 0.59 | 0.76 |
| Our | 47.1 (0.7) | **83.6** (0.2) | 0.58 | 0.77 |
| | MAGXLNET | | | |
| Base | 46.7 (0.4) | 83.9 (0.3) | 0.59 | 0.77 |
| Our | 47.1 (0.3) | 84.2 (0.3) | 0.57 | 0.77 |
| | MIM | | | |
| Base | 53.3 (0.5) | 79.1 (0.3) | 0.59 | 0.71 |
| Our | **54.1** (0.8) | **80.1** (0.2) | 0.57 | 0.73 |

Table 8: Results on sentiment analysis on a) CMU-MOSI and b) CMU-MOSEI. $Acc_7$, $Acc_2$ denote accuracy on 7, 2 classes respectively. $MAE$ is Mean Absolute Error and Corr is the Pearson correlation.

| | Accuracy $\uparrow$ | |
|---|---|---|
| Model | Base | Ours |
| LF | 71.4 (0.4) | 71.6 (0.4) |
| LFN | 71.1 (0.3) | **71.8** (0.3) |
| MFM | 71.4 (0.4) | **72.2** (0.6) |
| GB | 68.9 (0.6) | 69.3 (0.5) |
| Refnet | 70.6 (0.7) | **71.2** (0.5) |
| MFAS | 72.1 (0.5) | **72.5** (0.3) |

Table 9: Results on digit classification task with AVMNIST for various fusion architectures. The performance metric is Accuracy. Scores outside the average range of baseline models have been highlighted.

# E TECHNICAL ANALYSIS

Given a base model $\mathcal{F}$ with input $x = (x_i, x_2, ..x_k)$, we want to create an augmented model $\hat{\mathcal{F}}$ : $\mathcal{X} \times \mathbb{R}^d \to \mathcal{Y}$ with additional input $c \in \mathbb{R}^d$ such that $c = 0 \implies \hat{\mathcal{F}}(x, c) = \mathcal{F}(x)$. Recall that the function $\mathcal{F}$ mentioned in Section 2.1 is given by $\mathcal{F}(x) = P(F(G_1(x_1), G_2(x_2), ..G_K(x_K)))$.

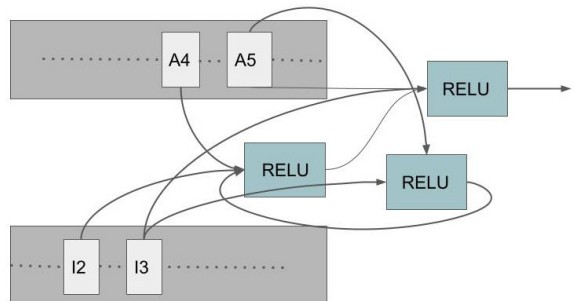

Figure 11: MFAS (Pérez-Rúa et al., 2019) based Multimodal Fusion Architecture for AVMNIST. Every arrow into the activation corresponds to a linear layer. The A4 and A5 represent the fourth and fifth layer of the audio encoder. Similarly I2 and I3 represent the second and third layer of the image encoder.

| Model | Base | Ours |
|--------|-------|-------|
| LFN | 1626k | 1655k |
| MFM | 1142k | 1163k |
| RefNet | 582k | 607k |
| GB | 659k | 671k |

Table 10: Parameter size comparison for models tested on AVMNIST

We create the desired network $\hat{\mathcal{F}}$ by providing $c$ to the unimodal feature generators $G_j$. We use the output of the fusion layer $F$ and project it back into the network as $c_t$ via the matrix/function $W_i$. Specifically we choose a modified generator $\hat{G}_i : \mathcal{X}_i \times \mathbb{R}^{d_i}$ to be given by $\hat{G}_i(x_i, c) = G_j([x_i, W_i(c)])$, where $W_i$ represents a matrix/network. This creates a recurrence relation which we unroll for $R$ steps. The final vector $c_R$ after $R$ steps serves as the output of fusion which is then provided to the predictor model $P$.

Mathematically, we can write the overall operation as :

$$\hat{G}_i(x_i, c_{t-1}) = G_j(x_i + W_i(c_{t-1})) \tag{7}$$

$$c_t = E(F(\hat{G}_1(x_1, c_{t-1}), .., \hat{G}_K(x_K, c_{t-1}))) \tag{8}$$

$$\hat{Y} = P(c_R) \tag{9}$$

$$\tag{10}$$

with the initial value $c_0 = \vec{0}$. We would like to draw the readers attention to the lack of $[t]$ subscript on the inputs $x$ in Equation 7 above. This is because the iterations on $t$ are on a dimension unrelated to any sequentiality in the input. Instead the model iteratively modifies its late-fusion features output, and makes it available to the unimodal network processing models/early stage features via the vector $c_t$. This is different from models like (Zadeh et al., 2018a) which use memory to store fusion vectors across different time-steps in the input, i.e. their networks processes $x_t$ to produce fusion output.

### E.1 LINEAR MODEL

Consider a simple linear model with multiple outputs and two input modalities. The two modalities are each $\mathbb{R}^D$ and the number of outputs is $K$. We consider the late and early fusion model of Figure 12 with only 1 layer between the input and fusion and from fusion to output. We take the fusion layer $F$ to be a concatenating operation. While the figure includes non-linearity, we will in this discussion look at the linear case.

**Analysis** The early fusion model works by concatenating the two inputs together, a linear transform to $\mathbb{R}^{2d}$ followed by another transform to $\mathbb{R}^K$. On the other hand the late fusion model corresponds to two modality wise transform to $\mathbb{R}^d$ which are then concatenated and transformed to $\mathbb{R}^K$. Note that in either case the second transformation is a $2d \times K$ matrix.

$$\begin{bmatrix} F_{11} & F_{12} \\ F_{21} & F_{22} \end{bmatrix} \begin{bmatrix} W_{11} & W_{12} \\ W_{21} & W_{22} \end{bmatrix} \begin{bmatrix} X_1 \\ X_2 \end{bmatrix} = \begin{bmatrix} F_{11}W_{11} + F_{12}W_{21} & F_{11}W_{12} + F_{12}W_{22} \\ F_{21}W_{11} + F_{21}W_{22} & F_{21}W_{12} + F_{22}W_{22} \end{bmatrix} \begin{bmatrix} X_1 \\ X_2 \end{bmatrix}$$

Late fusion on the other hand is expressible in the same formula by zeroing the off diagonal terms of $W$

$$\begin{bmatrix} F_{11} & F_{12} \\ F_{21} & F_{22} \end{bmatrix} \begin{bmatrix} W_{11} & 0 \\ 0 & W_{22} \end{bmatrix} \begin{bmatrix} X_1 \\ X_2 \end{bmatrix} = \begin{bmatrix} F_{11}W_{11} & F_{12}W_{22} \\ F_{21}W_{11} & F_{22}W_{22} \end{bmatrix} \begin{bmatrix} X_1 \\ X_2 \end{bmatrix}$$

First note that the rank of the effective matrix in early fusion is necessarily higher than that in late fusion. However even in cases when the rank of the effective matrix remains the same the matrix in late fusion is more constrained. To see this, consider a simple case where all $F_{ij}$ are diagonal matrices; in which case the ratio between the rows in the top-left quadrant (i.e. $F_{11}W_{11}$) is the same as the one in the bottom-left quadrant (i.e. $F_{21}W_{11}$) (and the same is true for the other quadrants as well). Similar constraints hold more generally. For example take the case of $d = 1, D = 2, K = 2$. The net transformation in this case is a 4x2 matrix of the form:

$$\begin{bmatrix} f_{11}w_{11} & f_{11}w_{12} & f_{21}w_{21} & f_{21}w_{22} \\ f_{12}w_{11} & f_{12}w_{12} & f_{22}w_{21} & f_{22}w_{22} \end{bmatrix}$$

One can see that the ratio of the elements of the first column and second column is the same. The same holds for third and fourth columns. Evidently not all rank two 4x2 matrices are of this form. As such there are functions in the early fusion variant which cannot be expressed in the late fusion design.

Next we compare this to the pro-fusion design presented in this work. We use a variant where the concatenated late vector is projected back as an input to the input encoders 1c, and the entire network is unrolled once.

The composite action of the backward connection plus unrolling is still linear and is given by the following composition

$$\begin{bmatrix} F_{11} & F_{12} \\ F_{21} & F_{22} \end{bmatrix} \begin{bmatrix} W_{11} + G_{11}W_{11} & G_{12}W_{22} \\ G_{21}W_{11} & W_{22} + G_{22}W_{22} \end{bmatrix} \begin{bmatrix} X_1 \\ X_2 \end{bmatrix}$$

The presence of off-diagonal entries which similar to early fusion breaks the structure imposed by late fusion in the earlier case. However this model is not as expressive as early fusion as there can be some dependencies between the matrix entries.

### E.2 MULTIPLICATIVE NONLINEARITY

While in the linear case, the extra freedom allowed by progressive fusion need not be very useful, the presence of multimodal interaction/ off-diagonal terms can have larger effects when dealing with non-linearity.

Consider a multiplicative non-linear layer $H : \mathbb{R}^D \rightarrow \mathbb{R}^d$ with the ability to provide any $d$ features obtained via pairwise multiplication of input features. For example for a vector input $[x_1, x_2, x_3, x_4]$ and output dimension $d = 3$, we can get any 3 of the 10 pairwise outputs i.e $[x_1^2, x_2^2, x_3^2, x_4^2, x_1x_2, x_1x_3, x_1x_4, x_2x_3, x_2x_4, x_3x_4]$. Using multiplicative non-linearity is useful for analysis as using the distributive property one can directly include behaviour of linear transformations. In Figure 12, we depict three simple non-linear fusion models with such non-linearity added in the layers. We denote the input features as $X^1, X^2$ for the two modalities, and their individual components are denoted by $X_1^1, X_2^1, X_1^2$ etc. We would also refer by $w^{p,q}$ the weight matrix applied on modality $p$ in the layer $q$ of the network.

**Analysis** After the first layer in the late-fusion design, the unimodal features are $\sum w_{ij}^{1,1} X_j^1$, $\sum w_{ij}^{2,1} X_j^2$ respectively. Then after the non-linearity, we get

$$\sum w_{ik}^{1,1} w_{il}^{1,1} X_k^1 X_l^1$$

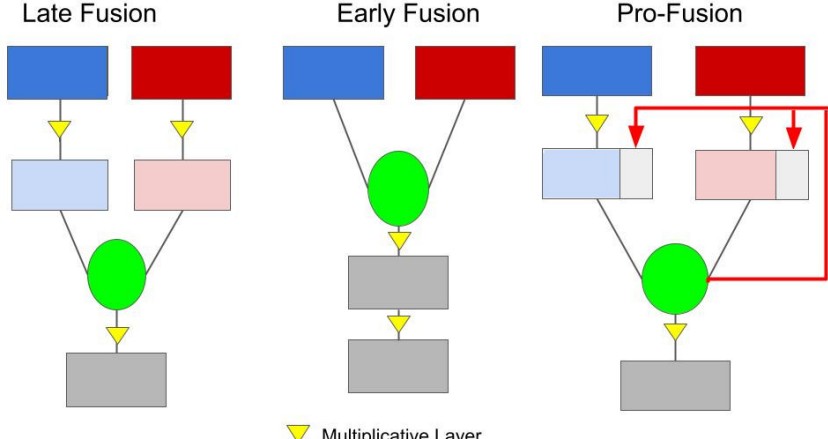

Figure 12: Representative Multimodal Fusion Architectures of Late fusion , Early fusion and Pro-fusion. The round green layer is the fusion layer (assumed to be concatenation). We also depict on the figure the location of multiplicative non-linearity with triangle, and highlight in red the back-projections of the pro-fusion design

These are then concatenated across modalities, and passed through another multiplicative non linearity. Hence the features obtained after this layer are given by

$$\sum w_{ik}^{1,1} w_{il}^{1,1} w_{i'm}^{2,1} w_{i'n}^{2,1} X_k^1 X_l^1 X_m^1 X_n^1$$

Effectively we have a linear combination of degree 4 terms that are symmetric in modalities i.e. $\{X_i^1 X_j^1 X_k^2 X_m^2, X_i^2 X_j^2 X_k^2 X_m^2, X_i^1 X_j^1 X_k^1 X_m^1\}$

However in the late fusion design the unimodal features are concatenated first, and then passed through the non-linearity. This produces after the first layer features of the form

$$w_{ij}^{1/2,1} X_i^{1/2} X_j^{1/2}$$

where $w^{1/2}$ and $X^{1/2}$ mean that choice over modalities 1 and 2 can be applied to $w$ and $X$ respectively. More simply one gets all pairwise terms from both modalities, instead of pairwise terms from only individual modalities. The next non-linearity, produces further multiplicative terms and we get linear combination of all degree 4 terms i.e. $X_i^{1/2} X_j^{1/2} X_k^{1/2} X_m^{1/2}$

Note here that no choice of linear operators between the layers can produce non-symmetric cross modal terms in the late fusion; and hence early fusion model has access to more features. On the other hand the first non-linearity in the early fusion case scales quadratically in the number of modalities. Specifically if the input dimensionality of each modality is $D$, and the number of modalities is $n$, then in early fusion the first non-linearity produces $\binom{nD}{2}$ feature outputs, whereas late fusion produces $n\binom{D}{2}$. Here $\binom{n}{k}$ refers to the binomial coefficient, also sometimes depicted as ${}^nC_k$ In the next non-linearity where a second multiplicative pairing occurs, early fusion needs $\binom{{}^{nD}C_2}{2}$ parameters whereas late fusion needs $\binom{n^{D}C_2}{2}$ . As such one needs many more samples to learn an early fusion model compared to a late fusion approach. The back-projections of Pro-fusion model however alleviates the lack of feature diversity in late fusion. The back-projections provides access to some cross-modal features before the fusion layer. This allows pro-fusion access to any given single asymmetric degree 4 term; however due to the limited dimensionality of the backprojected activations, not all combination of such degree 4 terms are accessible. For example if the backprojecting layer has size 2, then one can get only two pairs of independent asymmetric cross modal feature terms. Hence pro-fusion is more expressive pro-fusion but not as rich as early fusion.

# F   TRAINING BACKPROJECTION LAYERS

For training of backprojecting layers, we first build an augmented model $\hat{\mathcal{F}}$ by first extending the unimodal feature generators in the the base model. Next we add the backprojecting networks $W_i$ for each modalities. We pass the fused output to the unimodal generators through the backprojecting networks. Finally we fix a number of iterations, and unroll the model by applying the augmented network in a loop. The entire process is now differentiable, and autograd can compute the gradients.

```python
# base model is assumed given in the layers
class FusModel(nn.Module):
    def __init(unimodal_layers, fusion_layer, hp):
        self.unimodals = nn.ModuleList(*unimodal_layers)
        self.fusion_layer = fusion_layer
        self.num_modalities = len(unimodal_layers)

    def forward(inputs):
        uni_reps = [self.unimodals[_](inputs[_]) for _ in range(self.num_modalities)]
        fused_rep = self.fusion_layer(uni_reps)
        return fused_rep

class LateFusModel(nn.Module):
    def __init(unimodal_layers, fusion_layer, head, hp):
        self.base_model = FusModel(unimodal_layers, fusion_layer, head, hp)
        self.head = head
        self.num_modalities = len(unimodal_layers)

    def forward(inputs):
        fused_rep = self.base_model(inputs)
        return self.head(fused_rep)

class ProFusModel(nn.Module):
    def __init(num_modalities, unimodal_layers, fusion_layer, head, hp):
        self.back_projecting_layers = nn.ModuleList([self.build_back_layers(unimodal_layers[i
        ], fusion_layer.output_dim, hp) for i in num_modalities])
        self.augmented_readers = nn.ModuleList([self.extend_layer(unimodal_layers[i], hp) for
        i in  num_modalities])
        self.model = FusModel(self.augmented_readers, fusion_layer, hp)
        self.output_head = head
        self.hp = hp

    def forward(self, inputs):
        context_t = torch.zeros([inputs[0].shape[0], self.context_size])
        context_t = [self.back_projecting_layers(context_t) for _ in range(self.hp.
        num_modalities)]
        for _ in range(self.hp.num_unroll_steps):
            context_t = model_out = self.model(inputs, context_t)
            context_t = [self.back_projecting_layers(model_out) for _ in range(self.hp.
        num_modalities)]
    return self.output_head(model_out)

def train_epoch(model, optimizer, criterion, hp, train_loader):
    for i_batch, batch_data in enumerate(train_loader):
        modalities, tgt = batch_data
        optimizer.zero_grad()
        preds = model(modalities)
        loss = criterion(preds, tgt)
        loss += model.regularizer(hp)
        loss.backward()
        optimizer.step()
```

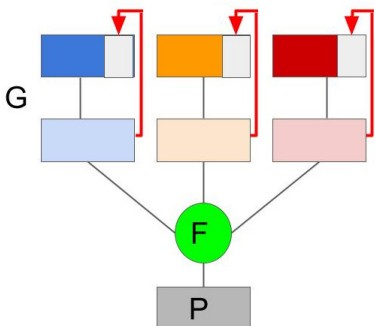

Figure 13: Example unimodal iterative network for ablation against ProFusion

## G   EXPLORING THE REPRESENTATIONS

Next, we analyze how the unimodal representations evolve over the unrolling steps. For this purpose, we consider the activations of unimodal networks $\hat{G}_j$ (equivalently, the inputs for the late fusion layer) as the unimodal representations. For these of experiments, we use LFN, MFM and Refnet models on AVMNIST. We train a linear classifier based on the unimodal representations from the training data and find its accuracy on the test data.

In Figure 6 we plot the relative test accuracy of both the audio and image features against the iteration number for all the models. We can see gains in all models after one step of unrolling. Since the individual modalities are quite incomplete on AVMNIST ($< 60\%$ accuracy on individual modalities) and the accuracy of only image modality at the first step is close to $60\%$; this increase is suggestive of greater information integration. Along with the overall trend, this suggests that *the model is incorporating more multimodal information in each unimodal representation.*