# OpenReview forum: "Progressive Fusion for Multimodal Integration"
_ICLR.cc/2024/Conference — ICLR 2024 Conference Withdrawn Submission_

### Official Review · Reviewer_mKqA · 2023-10-26

**Soundness:** 3 good
**Presentation:** 2 fair
**Contribution:** 2 fair
**Rating:** 5
**Confidence:** 3

**Summary:**

This paper presents an iterative representation refinement approach called Progressive Fusion for multimodal integration. It aims to solve the drawbacks of early fusion and late fusion. For this purpose, it uses backward connections which connect the late fused representation to unimodal feature generators, thus providing cross-modal information to the early layers.

**Strengths:**

1. The motivation for the method is clear. The differences and improvements between the proposed progressive fusion framework and existing early and late fusion frameworks are explicit.
2. It conducts a variety of tasks to validate the performance improvements, including sentiment prediction, multimedia classification and financial time series prediction.
3. As for the experiment results, the results of the proposed method show superiority.
4. The supplementary material is rich, providing a lot of supplements and support to the main text.

**Weaknesses:**

1. The article devotes a significant amount of time to introducing the motivation behind the method and its differences from existing fusion frameworks. For example, Sec. 3.1 and the previous introduction are repetitive. However, the introduction to the method is brief.
2. The proposed framework feeds the late fused representation to the early unimodal feature generator. However, in the early fusion, the unimodal feature generators may be bottlenecked or not powerful enough (required information might be lost or compressed too much to be recovered correctly). If the late fused representation is bottlenecked or not powerful enough, it seems unreasonable and effective to input it into the front end to provide information.
3. The competitors are methods proposed in 2017-2021. It lacks a comparison with some state-of-the-art methods.
4. The experiment lacks schematic diagrams and qualitative results comparison.

**Questions:**

1. This framework forms a closed loop between input and output in the form of feedback, why can the training of the network still be carried out?
2. The logical relationship in Fig. 2 is unclear. $X_1$ and $X_2$ are known multimodal information that needs to be fused to obtain the results. However, the directions of the arrows and the directions of information transmission contradict each other, and the source of Z and its relationship with input and output are not clear.

---

### Official Review · Reviewer_FVUs · 2023-10-29

**Soundness:** 3 good
**Presentation:** 3 good
**Contribution:** 2 fair
**Rating:** 3
**Confidence:** 3

**Summary:**

This paper focus on enhancing multimodal fusion which is one of the fundamental problem in multimodal learning. The authors dementrate that current multimodal fusion methods either run the risk of information loss in the respective unimodal or suffer from various challenges like feature shifts, cross modal distributional changes, and so on. Thus, this paper introduces a new fusion method called Progressive Fusion to conbining the benefits of both early and late fusion, which fusing multimodal information in an iterative manner.

**Strengths:**

1. The paper introduces a novel cyclic fusion method, improving the expressiveness of the representations. The proposed method is novel and well-motivated to me.

2. The paper proposes an interesting issue in multimodal fusion i.e., deep unfolding/iterative models that do not cycle cross-modal information and still suffer from the “fuse it or lose it”. I believe this issue is worth to be well addressed.

3. The paper is well written and follows a good structure.

**Weaknesses:**

1. **Related works**: In the introduction section, only two types of model fusion are mentioned, early fusion and late fusion, without including intermediate fusion. Therefore, the first contribution might not be strong enough for me, as various intermediate fusion methods might already encompass this aspect. In subsequent experiments, it's also recommended to include comparative experiments with relevant intermediate fusion methods e.g. concat, self-attention and so on. Additionally, late fusion should fuse information at the decision level rather than the feature level, which would be more appropriate in my view.

2. **Impact of potential noise**: In my opinion, When one modality is noised, the proposed method fuses features into a single modality, potentially introducing additional noise into the clean modality during cyclic progress.

3. **Experimental comparisons**: It's mentioned that the proposed method's parameters are consistent with those of the comparative methods. However, the supplementary material states that the increase in the number of parameters is minimal. In comparison to the original model, the model parameters increase by almost 5% in RefNet and nearly 2% in GB, LFN, and MFM methods. Moreover, according to Figure 5, most of the model performance improvements proposed in your paper are below 1%. Therefore, it's advisable to conduct more ablation experiments. Additionally, the phenomenon in Figure 6 seems a little trivial in my view because unrolling provides one modality with information from another, leading to an increase in accuracy. This is because, at this point, single-modality classification is not purely based on one modality's information.

4. **Other Issues**: There is a format issue in Table 6 of the supplementary material. The definition of "ROBUTSNESS" in the supplementary material and the main text doesn't seem consistent: one is presented in terms of negative MSE, and the other resembles a positive metric like accuracy. Please ensure clarity in graphical representations and labels.

**Questions:**

Please refer to weaknesses.

---

### Official Review · Reviewer_qtxk · 2023-10-31

**Soundness:** 3 good
**Presentation:** 3 good
**Contribution:** 2 fair
**Rating:** 5
**Confidence:** 4

**Summary:**

This paper presents Progressive Fusion, a novel approach to multimodal integration that combines early and late fusion techniques to improve the performance of machine learning models. The authors evaluate their approach on several multimodal datasets and demonstrate consistent improvements in accuracy and robustness compared to other fusion methods.

**Strengths:**

1.The Progressive Fusion strategy adeptly amalgamates the merits of both early and late fusion paradigms, thus culminating in an enhanced efficacy across diverse multimodal ventures.
2.Demonstrating intrinsic adaptability, the methodology can be seamlessly integrated into an array of machine learning architectures and modality spectrums.
3.The rigorous empirical assessments conducted across varied datasets not only attest to the approach's robustness but are also supplemented by meticulous delineations of the procedural mechanics and experimental configurations, thereby facilitating reproducibility.

**Weaknesses:**

1.The manuscript, while delving deep into the technical facets of Progressive Fusion, offers limited discussion on its real-world applications and broader implications. Moreover, its empirical evaluation seems restricted, being limited to a narrow array of datasets, raising concerns about the universality of the approach.
2.A detailed comparative analysis with other leading multimodal integration techniques is noticeably missing, potentially overlooking nuances and advancements made by contemporaneous methods.
3.The presentation and accessibility of the content may deter a broader readership, as results are primarily tabulated and the discourse assumes substantial technical proficiency.
4. The work lacks substantial innovation, primarily leveraging a now prevalent progressive network structure.

**Questions:**

1.Comparative Analysis with Existing Multimodal Integration Techniques: With the burgeoning landscape of multimodal integration methods, it becomes imperative to benchmark Progressive Fusion against contemporaneous state-of-the-art techniques. Specifically, how does Progressive Fusion fare when juxtaposed against other leading approaches across a diverse set of tasks, encompassing varying complexities and data types? Such a comparative analysis is pivotal to discern the unique advantages and potential shortcomings of the method.
2.Applicability in Real-world Multimodal Systems: While the theoretical underpinnings of Progressive Fusion have been elucidated, its pragmatic applications remain an area of intrigue. For instance, can Progressive Fusion be seamlessly integrated into affective sentiment detection systems that amalgamate auditory, textual, and visual cues? Similarly, is it adept at facilitating sophisticated multimedia analysis tools that demand concurrent processing of disparate modalities? These questions beckon exploration to gauge the method's real-world relevance and versatility.
3.Versatility Across Modalities and Scalability Concerns: The performance metrics of Progressive Fusion across standard modalities, such as audio, video, and text, remain a subject of inquiry. How does the method adapt and perform when confronted with unconventional or less-explored modalities? Additionally, as we transition to an era of big data, the scalability of any method becomes paramount. Thus, is Progressive Fusion inherently equipped to handle vast datasets without compromising on efficiency? And can it gracefully scale to cater to more intricate model architectures?
4.Future Extensions and Incorporation of Diverse Data Types: Progressive Fusion's modular structure potentially lends itself to extensibility. Yet, how might it be augmented to assimilate additional modalities or unconventional data types? Exploring such extensions could pave the way for a more holistic and inclusive multimodal integration approach.

---

### Official Review · Reviewer_HPuP · 2023-11-01

**Soundness:** 1 poor
**Presentation:** 2 fair
**Contribution:** 2 fair
**Rating:** 5
**Confidence:** 5

**Summary:**

This paper proposes an iterative representation refinement approach called Progressive Fusion for better integration of multimodal information from various sources. Extensive experiments on several tasks demonstrate the effectiveness of the proposed model.

**Strengths:**

1. The paper is clearly written and contains sufficient details and thorough descriptions of the experimental design.
2. Extensive experiments are conducted to verify the effectiveness of the proposed method.

**Weaknesses:**

1. While the authors provide a motivation example in Section 3.1, the motivation and analysis are purely built on hypothesis. This makes the motivation less convincing .

2. For the table 1-3, the choices of baselines are quite weak. In fact, there are numerous advanced baselines, such as ViLT, ALBEF and METER, which focus on how to fuse different modalities. These baselines should be included in the comparison.

**Questions:**

See the above weakness. The main concerns are that the model is not well-motivated and baselines are quite weak to support the claims.